# Dysbiosis in Inflammatory Bowel Disease: Pathogenic Role and Potential Therapeutic Targets

**DOI:** 10.3390/ijms23073464

**Published:** 2022-03-23

**Authors:** Patricia Teixeira Santana, Siane Lopes Bittencourt Rosas, Beatriz Elias Ribeiro, Ygor Marinho, Heitor S. P. de Souza

**Affiliations:** 1Department of Clinical Medicine, Federal University of Rio de Janeiro, Rio de Janeiro 21941-913, RJ, Brazil; pattsant@gmail.com (P.T.S.); sianeros@gmail.com (S.L.B.R.); bakerribeiro@gmail.com (B.E.R.); ygormarinho@rocketmail.com (Y.M.); 2D’Or Institute for Research and Education (IDOR), Rua Diniz Cordeiro 30, Botafogo, Rio de Janeiro 22281-100, RJ, Brazil

**Keywords:** inflammatory bowel disease, gut dysbiosis, inflammation, immunomodulation, epigenetics

## Abstract

Microbe–host communication is essential to maintain vital functions of a healthy host, and its disruption has been associated with several diseases, including Crohn’s disease and ulcerative colitis, the two major forms of inflammatory bowel disease (IBD). Although individual members of the intestinal microbiota have been associated with experimental IBD, identifying microorganisms that affect disease susceptibility and phenotypes in humans remains a considerable challenge. Currently, the lack of a definition between what is healthy and what is a dysbiotic gut microbiome limits research. Nevertheless, although clear proof-of-concept of causality is still lacking, there is an increasingly evident need to understand the microbial basis of IBD at the microbial strain, genomic, epigenomic, and functional levels and in specific clinical contexts. Recent information on the role of diet and novel environmental risk factors affecting the gut microbiome has direct implications for the immune response that impacts the development of IBD. The complexity of IBD pathogenesis, involving multiple distinct elements, suggests the need for an integrative approach, likely utilizing computational modeling of molecular datasets to identify more specific therapeutic targets.

## 1. Introduction

In the last two decades, the gut microbiota has become a focus of major interest in the study of inflammatory bowel disease (IBD) pathogenesis. Technological advancements have allowed the characterization of several gut microbiome abnormalities in patients with Crohn’s disease (CD) and ulcerative colitis (UC), the two major forms of IBD. Abnormal immune reactivity against commensal microorganisms [1,2] and defects in innate and adaptive immunity have long been described in studies on IBD [3,4,5]. Nonetheless, the most consistent association between IBD and bacteria has been derived from animal models. For example, germ-free mice do not develop colitis, and inflammatory changes can be induced after colonization with commensal bacteria [6]. In a study using interleukin (IL)-10-deficient mice, which are genetically susceptible to colitis development, antibiotic administration early in life increased the risk of colitis [7].

Regarding genetic predisposition, several studies have identified altered regulators of the complex network underlying IBD in patients, many of which control the immune response to microbes. Nucleotide-binding oligomerization domain 2 (NOD2) polymorphisms, which encode an intracellular pattern recognition receptor and regulate the production of defensins by Paneth cells, have been associated with the risk of CD [8]. Several other gene variants related to bacterial clearance or protection against epithelial invasion have also been associated with IBD [9]. Currently, healthy immune homeostasis is linked to a state of tolerance towards resident microbiota, and disequilibrium of normal homeostatic conditions has been proposed as a necessary event for the initiation and/or maintenance of inflammation in IBD [10].

Although there is not yet a clear definition of what a healthy gut microbiome is, dysbiosis has generally been defined as an altered balance between the microbiota and its host [11]. Hence, commensal microbiota play a critical role in the early education of the immune system and microorganisms, and their products modulate immune responses through the induction of immune cells, signaling pathways, and inflammatory mediators [9]. In contrast, exposure to environmental triggers, such as dietary components, gastrointestinal infections, medications, psychological stress, and smoking, in genetically susceptible individuals leads to dysbiosis-associated mucosal immune dysfunction, which delineates IBD. Prolonged dysbiotic conditions, characterized by increased aggressive bacterial strains and decreased regulatory species, result in dysfunction of the mucosal immune response (Figure 1). Together with defective intestinal barrier function, gut dysbiosis is likely to sustain mucosal inflammation and potentially lead to IBD [12]. In this review, we aimed to discuss and summarize the basic mechanisms and potential associations between dysbiosis and IBD development, as well as exposome influences, methodological approaches to the microbiome, epigenetic changes, and recent developments in therapeutics.

## 2. Intestinal Microbial Dysbiosis

The gut microbiota is an important physical, chemical, and immunological interface between the environment and host; thus, any dysregulation or breakdown of this barrier can contribute to disease states. For example, altered physical epithelial barrier function, a thinner mucus layer, and altered responses to endoplasmic reticulum stress (via mutations in *MUC19*, *ITLN1*, *FUT2*, and *XBP1*) have all been identified as risk factors for IBD [13,14,15]. Currently, the pathogenesis of human IBD is believed to involve inappropriate activation of the immune system when genetically susceptible individuals are exposed to gut antigens, such as microbiome components [16]. Although alterations in the gut microbiome have been proposed to be critical in IBD pathogenesis, it is not yet clear how this process occurs and whether dysbiosis is a central cause or a common consequence of the disease [17].

In healthy individuals, 99% of gut bacterial phyla are Firmicutes, Bacteroidetes, Proteobacteria, and Actinobacteria. Firmicutes and Bacteroidetes account for approximately 90% of the total microbiome composition. These phyla are critically important in maintaining gut homeostasis and produce short-chain fatty acids (SCFAs), especially butyrate and propionate, from the fermentation of dietary components such as indigestible fibers. SCFAs are important energy sources for colonic mucosa cells but have also been shown to play key roles in regulating immune homeostasis [18]. Dysbiosis is defined as an alteration in gut microbiota composition and diversity and a shift in the balance between commensal and potentially pathogenic microorganisms [19]. Several pieces of evidence support the role of the microbiome and dysbiosis in IBD development. For example, experimental mice subjected to germ-free conditions develop attenuated colitis [20]. In studies using mouse models, the transfer of bacterial strains associated with IBD induces intestinal inflammation in genetically susceptible mice [21]. Similarly, fecal transplantation from human IBD donors to germ-free mice stimulates proinflammatory responses, with increased Th17 cell infiltration and proinflammatory mediators compared with transplants from healthy human donors [22]. Britton et al. colonized groups of adult wild-type or *Rag1*-deficient mice in germ-free conditions with human microbiota and assessed the mucosal immune response. Microbiota from healthy human donors induced, on average, higher frequencies of RORγt + Foxp3 + Treg cells in the intestinal lamina propria and prevented disease exacerbation. In contrast, microbiota from IBD donors resulted in enhanced RORγt + Th17 effector cell frequencies and enhanced disease severity in colitis-susceptible mice [23].

Determining the groups of microbes that are related to the development of intestinal inflammation has been a focus of extensive research. Patients with IBD tend to present several changes, not only in composition, but also in the diversity of their microbiome populations when compared to healthy individuals (Table 1). Evidence shows that alterations in microbiome components can also be involved in different IBD phenotypes [24]. The IBD microbiota has been characterized by an increase in the abundance of Bacteroidetes and Proteobacteria and a decrease in Firmicutes compared to control individuals. Specifically, levels of *Faecalibacterium prausnitzii*, a highly metabolically active commensal bacterium, are reduced in individuals with IBD [25]. Patients with IBD have reduced microbiome diversity (mostly a decrease in the relative abundance of Firmicutes) and an increase in the presence of Proteobacteria, such as *Enterobacteriaceae* and *Bilophila*, and certain members of Bacteroidetes [26]. Dysbiosis can potentially lead to a reduction in key functions necessary for maintaining intestinal barrier integrity and gut homeostasis. Therefore, alterations in the immune response and proinflammatory activity could be due to a dysbiotic microenvironment.

Clooney et al. showed that among the microbial species found to be significantly increased in CD compared to controls, there was an increased presence of *Ruminococcus gnavus* and *Fusobacterium nucleatum*. Conversely, the presence of *Ruminococcus albus*, *Eubacterium rectale*, and *Faecalibacterium prausnitzii* were decreased in CD. *Eubacterium* and *Roseburia* were among the most important species in classifying either CD or UC compared with controls [24,47]. Furthermore, some species are particularly associated with certain subgroups of patients. For instance, *Bacteroides vulgatus*, *Akkermansia muciniphila,* and *Escherichia/Shigella* were increased in patients with a history of prior surgical resection [44]. In the largest pediatric Crohn’s cohort to date, including >400 patients and 200 controls, microbiomes from new-onset CD cases in multiple gastrointestinal locations were analyzed by Gevers et al. An axis defined by an increased abundance of bacteria such as *Enterobacteriaceae, Pasteurellacaea*, and *Fusobacteriaceae* and decreased abundance of *Erysipelotrichales, Bacteroidales*, and *Clostridiales* was strongly correlated with disease status. Moreover, microbiome comparison between patients with CD with and without antibiotic exposure indicated that antibiotic use amplified the microbial dysbiosis associated with CD in a new-onset pediatric cohort. The microbial dysbiosis index, which is characterized by the differential relative abundance of specific taxa, was associated with disease severity. Additionally, the rectal mucosa-associated microbiome, but not the fecal microbiome, has been shown to be a robust disease predictor [30]. Similarly, another study of a pediatric IBD cohort showed significant correlation between microbiota composition and disease severity, with resolution of dysbiosis in patients responding to anti-tumor necrosis factor (TNF) therapy [48]. Although changes in the gut microbiota profile in new-onset and treatment-naive pediatric patients with IBD were further corroborated in a recent systematic review, no clear conclusion can be drawn at the moment due to inconsistent results and heterogeneous methodologies [49].

In one of the largest longitudinal analyses of the IBD microbiome, Halfvarson et al. dissected the long-term dynamic behavior of the gut microbiome by comparing patients with IBD with healthy controls. Phylogenetic analysis of fecal samples showed that gut microbiomes of different IBD subtypes displayed different species distributions relative to controls. They identified potential microbial indicators of IBD subtypes, including genera such as *Lachnospira*, *Clostridium*, *Oscillospira*, and many unidentified *Ruminococcaceae.* Furthermore, they found that the microbiomes of patients with IBD fluctuate more than those of healthy individuals, based on deviation from a newly defined healthy plane. In addition, patients with ileal CD deviated the most from the healthy plane, especially those who underwent surgical resection. Interestingly, the microbiomes of some patients with IBD periodically visited the healthy plane and then deviated away from it [17]. In a study of 132 participants with IBD and controls, Lloyd-Price et al. assessed host and microbial data from colon biopsies, blood, and stool, for one year each. Principal coordinate analysis based on species-level Bray–Curtis dissimilarity showed that most variation was driven by a trade-off between the phyla Bacteroidetes and Firmicutes. Samples from individuals with IBD (particularly CD) had lower alpha diversity. Moreover, in patients with CD, taxonomic perturbations during dysbiosis were observed, such as a depletion of obligate anaerobes, including *F. prausnitzii* and *Roseburia hominis*, and an enrichment of facultative anaerobes such as *Escherichia coli*. In the metabolome, SCFAs were generally reduced during IBD dysbiosis. Overall, metabolite pools were less diverse in individuals with IBD, paralleling the observations of microbial diversity [50]. The investigation of whether the response to treatment with biologic agents could be associated with alterations in the composition of the intestinal microbiota of patients with CD was performed in a prospective study. The therapeutic intervention based on adalimumab was associated with the restoration of a eubiotic environment after six months of treatment. Particularly, in the cohort of patients with CD, those receiving adalimumab displayed a reduction in *Proteobacteria* and an increase in *Lachnospiraceae.* These results characteristically predominated among those patients who achieved therapeutic success, suggesting that dysbiosis could be directly involved with the response to treatment [51].

The role of pathogenic components in IBD has also been studied. For example, adherent-invasive *E. coli* (AIEC) is more prevalent in the mucosa of patients with IBD than in healthy individuals. Overall, the prevalence of AIEC in the mucosa of adult patients with CD ranges from 21–63%, and it is more associated with ileal CD than with colonic disease [41]. *Mycobacterium avium*, especially the subspecies *paratuberculosis*, has also been implicated in IBD pathogenesis. The abundance of these bacteria is higher in patients with IBD than in controls, and they are associated with increased production of proinflammatory cytokines [31]. Similarly, *Listeria monocytogenes*, which induces a Th1-type immune response [52], has also been shown to be increased in patients with IBD [33].

Archaea members have also been identified as part of the human microbiome, especially methane-producing species, such as *Methanobrevibacter smithii*, *Methanosphaera stadtmanae*, and *Methanomassiliicoccus luminyensis* in human stools, and *Methanobrevibacter oralis* in the oral mucosa [53,54]. Dysbiosis appears to affect the relative abundance of archaea members and, particularly in patients with IBD, *M. smithii* had a reduced abundance, whereas the immunogenic *M. stadtmanae* was remarkably increased compared to healthy controls [55,56].

Viruses and fungi are widely present in the gut and may play important roles in homeostasis. Changes in the enteric group of viruses in the gastrointestinal tract can have consequences on the bacterial microbiome and its diversity, as viruses are drivers of bacterial resistance. For instance, viruses can be responsible for the horizontal transfer of genetic material among bacterial communities, which implies a change in the balance of different ecosystems. An increase in the abundance of *Caudovirales* bacteriophages has been observed in patients with CD [57,58]. In addition, results from another study demonstrated an increased abundance of phages infecting Clostridiales, Alteromonadales, and *Clostridium acetobutylicum*, as well as viruses from the Retroviridae family, in patients with IBD [59,60].

Similarly, fungal components regulate and trigger immune responses. However, the gut mycobiome is less stable than the bacterial microbiome; therefore, a mycobiome signature has not yet been described [61]. *Candida* is the most prominent component of the fungal microbiota in humans. Its cell wall constituents, such as beta-glucans, chitin, and mannoses, can activate components of the innate immune system, such as Toll-like receptors (TLRs) 2 and 4, dectin-1, CD5, CD36, and SCARF1, and complement system components. The activation of these molecules leads to immune signaling and proinflammatory responses. Some studies have proposed a correlation between the gut mycobiome and gut bacterial components. Mason et al. showed that the colonization of *C. albicans* strain CHN1 in the stomach and cecum of C57BL6 mice prompted an overgrowth of *Lactobacillus spp*. and *Enterococcus spp*. during antibiotic treatment [62]. However, Bernardes et al. demonstrated that bacteria may influence fungal colonization. They showed that the colonization of bacteria together with fungi increased the relative abundance of *C. parapsilosis* and *Issatchenkia orientalis*, and a lack of co-colonization with bacteria or elimination of bacteria by antibiotics led to an overgrowth of *C. albicans* [63].

## 3. Influence of the Exposome

The first organisms on earth were microbes, and they have evolved and adapted to live in extreme environments all over the planet. All biological entities appearing later on Earth, including mammals, have evolved in a microbially dominant world. Therefore, humans have coevolved with microbes immersed within two complex ecological communities: the external and internal microbial environments. In this dualistic world of microbes, the exposome and gut microbiome impact each other as well as all other “–omes” in a reciprocal manner [64,65].

### 3.1. Geosocial Factors

Environmental factors have been associated with complex health conditions, including chronic immune-mediated inflammatory diseases (IMIDs), which have been increasing in incidence during the last century. Features are shared among the different IMIDs, including IBD, such as an inflammatory basis, multifactorial nature, and yet-unknown causes. In addition, epidemiological data revealing the co-occurrence of IMIDs and geographic expansion reinforce a common pathophysiological background that has evolved over the past several decades to reach a worldwide distribution [66]. The emergence of IMIDs and IBD has been linked to societal transformations, most commonly socioeconomic development or industrialization [67,68]. Such changes have been almost invariably accompanied by increasing urbanization [69], which, in turn, has been associated with distinct gut microbiota compositions different to those found in rural areas that are supposedly protective against IBD development [70,71].

Socioeconomic development and social behavior are crucial elements fueling the emergence of IBD [72]. Such changes have been associated with improvements in sanitation, quality of water supply in distribution systems, and a resulting decrease in infectious diseases, which constitute the basis of the hygiene hypothesis [73]. Nonetheless, these changes bring about several other simultaneous environmental modifications that need to be considered. It is important to highlight, for example, changes in homes, family structures, workplaces, dietary habits, the widespread use and production of chemicals, and the use of medications, including antibiotics. Growing urbanization has led to a continuous increase in population density in cities that have become progressively more polluted, competitive, and stressful, causing dramatic changes in peoples’ lifestyles. Moreover, the attraction between manpower and industry, or other economic activities, resulted in more human agglomeration, whether that be in households or factories; it also gathered people from different backgrounds, be they genetic, geographic, or cultural. These observations are in accordance with previous studies showing that individuals who migrate from low to high prevalence IBD areas, that is, from less developed to more developed areas, are more susceptible to developing IBD, predominantly affecting the first- and second-generation offspring of these immigrants [74,75,76,77]. In addition, among the potentially relevant stimuli from the exposome, cohabitation has been shown to strongly affect immune responses [78]. Transmission of microbial strains, predominantly detected among first-degree relatives sharing a household, has recently been demonstrated [79], helping to explain the link between the exposome and the immune response. This also reinforces a microbial basis for IMIDs.

While several human diseases have been associated with abnormalities in host-associated microbial communities, and the human body is seen as an ecosystem [80], defining a healthy microbiome continues to represent a complex challenge due to the formidable variability shown in population-based studies [81,82]. This is also true for IBD, as a large study confirmed a reduction in microbial diversity in patients with CD and UC but did not explain the increased variability compared to controls [24]. Whether such a variance is stochastic or due to environmental factors has not yet been established [83]. Nevertheless, the microbiome reflects a complex combination of endogenous and exogenous elements, particularly environmental and lifestyle factors. Previous studies have shown that the gut microbiome of Western populations is characteristically less diverse [84,85,86]. As the intestine represents the largest surface of contact with the external environment, IBD could be facilitated by a combination of both a cleaner external milieu (as in the hygiene hypothesis) and an impoverished biome influencing the internal milieu to become less diverse, resulting in inappropriate immune system education and responses.

The hypothesis that loss of biodiversity is an important environmental factor has been supported by data showing that reduced contact of people with the natural environment may negatively impact the commensal microbiota and its immunomodulatory properties [87,88]. In addition to its worldwide distribution and progressive increase as a result of diverse human activities, loss of biodiversity has been regarded as a critical factor in the rise of allergic diseases [89], among which asthma, in particular, has been intimately associated with IBD [90]. Loss of biodiversity has recently been proposed as a novel factor in the pathogenesis and prevention of IBD, based on the non-uniform disease distribution in large developing countries, showing pronounced regional dissimilarities and disease hotspots associated with specific geosocial and ecosystem factors [91].

### 3.2. Antibiotics

Exposure to antibiotics has been associated with increased risk for developing IBD, especially CD [92,93]. Evidence from different studies has shown that patients diagnosed with IBD during childhood were more likely to have been exposed to antibiotics early in life [94]. In a pediatric prospective study, the strongest association between antibiotic use and future development of IBD was in the first 3 months following the use of antibiotics and among children who had more courses of antibiotics [95]. Contrarily, a recent study found that exposure to antibiotics during pregnancy, but not in infancy, is associated with an increased risk of early onset IBD [96]. Although current evidence does not confirm a consistent causal link with IBD, early exposure to antibiotics has been suggested to affect the development of tolerance to the gut microbiota, consequently raising inappropriate immune reactivity that underlies chronic intestinal inflammation [97]. Additionally, recent evidence from a study investigating the microbiome of humans, domestic animals, and their environment, in relation to antibiotic use, suggested the exchange of antimicrobial-resistant strains between reservoirs [98]. Together, these data appear to support the idea that the risk of developing IBD associated with intestinal dysbiosis may occur at both the individual and community levels. This also includes crosstalk with nonhuman components, reinforcing the existence of dynamic interactions between the environment and host regarding the exchange and sharing of microorganisms.

### 3.3. Dietary Factors

Several studies have investigated diet, arguably the most ubiquitous environmental factor, and its potential to shape the gut microbiota. For instance, evidence has shown that a high-calorie diet, consisting of fat- and carbohydrate-based foods, determines a preferential expansion of the genera *Bacteroides* and *Prevotella* and the Bacteroidetes phylum in adults, with shifts occurring in a relatively rapid fashion [99]. In another study, strictly animal-based food increased the relative abundance of bile-tolerant microorganisms, reducing the presence of microorganisms capable of metabolizing dietary plant polysaccharides. These results showed shifts between carbohydrate and protein fermentation, confirming that the microbiota can rapidly adjust to changes in dietary patterns. Moreover, changes in microbial composition were followed by changes in the molecular output of the microbiome with dietary interventions. SCFAs, products of bacterial digestion of fibers with critical homeostatic functions in the mucosa and anti-inflammatory properties, were shown to increase with plant-based diets. This may explain why a reduction in SCFAs in a typical Western-style diet (animal-based, high-calorie, high-fat, and low-dietary fiber) has been associated with the risk of IBD [100]. In fact, a Western-style diet, rich in sugar and fat, has been the predominant profile associated with a higher risk of developing IBD. While individuals who consume higher proportions of red meat and fats have a higher risk of IBD, others who predominantly consume fibers from vegetables and fruits have a lower risk [101,102]. Regarding dietary fat content, particularly polyunsaturated fatty acids, recent data indicate that a high omega-6 to omega-3 ratio, typical of Western-style diets, is associated with proinflammatory effects [103]. Furthermore, polyunsaturated fatty acids have been shown to exert not only effects on the immune response, directly acting on immune cells, but also influence the composition of the gut microbiota, thereby affecting host–microbiome interactions at different levels [104]. The consumption of processed foods, usually low in omega-3 fatty acids and micronutrients such as zinc and vitamins D and E, another common feature of Westernized diets, has also been associated with the development of chronic inflammatory diseases [105,106,107,108]. Globally, major shifts in dietary patterns towards progressively more Westernized diets, together with socioeconomic and demographic changes, represent a global transition that may explain the widespread increase in the rates of several metabolic and IMIDs [109], potentially involving changes in the gut microbiome and its interaction with the host.

## 4. Genetic Susceptibility

A clear connection with genetic predisposition has long been demonstrated in IBD, more so in CD than in UC [110]. There are over 200 genetic loci associated with IBD susceptibility, most of which regulate host–microbe interactions and immune-related pathways [110,111]. Some of the more studied genes include those involved in IL-23 receptors and Janus-activated kinase signaling, and those in innate mucosal defense, cytokine production, lymphocyte activation, epithelial barrier integrity, and multiple proteins involved in autophagy [112]. Genome-wide association studies have highlighted higher IBD genetic risk in individuals with NOD2 receptors, autophagy-related protein 16-like 1 (*ATG16L1*), immunity-related GTPase family, M (*IRGM*), IL-23 receptor gene, protein tyrosine phosphatase, non-receptor type 2 (*PTPN2*), X-box binding protein 1 (*XBP1*), and leucine-rich repeat kinase 2 (*LRRK2*) variants [111,113]. Genetic risk variants are also associated with changes in microbiota composition; for example, *Roseburia* spp., an acetate-to-butyrate converter, was less abundant in patients with IBD with these high-risk mutations [114].

Mutations in autophagy-related genes alter anti-bacterial, fungal, and viral responses and impair the clearance of various invading pathogens such as *Mycobacterium tuberculosis*, Group A Streptococcus, *L. monocytogenes*, and *E. coli* [115,116,117]. An *ATG16L1* single nucleotide polymorphism (SNP) confers susceptibility to CD and is a common genetic variation present in 40–50% of the population, although most individuals with this SNP do not develop IBD [118,119]. The role of autophagy variants in *Salmonella* clearance is not well established. Messer et al. observed that *ATG16L1* deficiency promoted cell resistance to *Salmonella*, while Conway et al. observed autophagy induction after *Salmonella* infection with the participation of ATG16L1 in intestines [120,121]. These variants also affect antimicrobial peptide production by Paneth cells, cytokine production, antigen presentation, and response to endoplasmic reticulum stress [122]. *Atg16L1*-deficient mice exhibited elevated inflammasome activation and IL-1β production when stimulated with lipopolysaccharides (LPS) and abnormalities in ileal Paneth cells, such as the escape of antimicrobial peptides into the cytoplasm [123]. There is crosstalk between NOD2 and ATG16L1, as NOD2 activation triggers autophagy in dendritic cells with the participation of ATG16L1, and deficiency in ATG16L1 heightens cytokine production via NOD [124,125]. Patients with CD with risk variants of *ATG16L1* or *NOD2* present with abnormal Paneth cell morphology [126]. In mouse models, decreased expression levels of *Atg5*, *Atg7*, or *Atg4B* generated abnormal Paneth cell functions, and in CD-like ileitis, deficiency of *Atg16L1* also altered Paneth cell morphology [127]. In addition, norovirus infection in *Atg16L1*-deficient animals increased their susceptibility to dextran sodium sulfate (DSS) in a TNF-dependent phenotype resembling aspects of IBD [58]. Complete knockout of *Atg3*, *Atg5*, *Atg7*, or *Atg16L1* is lethal in mice, and impairment of either *Atg7* or *Atg16L1* results in severe CD-like transmural ileitis [128]. Autophagic defects also worsen goblet cell function, production of mucus membrane defenses, and absorptive functions of the microvilli [129].

NOD2 recognizes bacterial peptidoglycan (muramyl dipeptide) in the cell walls of Gram-negative and Gram-positive bacteria and triggers the production of intestinal antimicrobial peptides to protect cells and immune responses in the gut [130]. NOD2 activation leads to NF-κB activation and production of IL-1b, TNF-α, IL-6, IL-8, and α-defensins [130,131]. NOD2 interacts with autophagy-related proteins to help destroy intracellular pathogens, and mutations in *NOD2*, also known as the caspase recruitment domain family, member 15 gene (*CARD15*), disrupt Paneth cells’ ability to recognize and eliminate invading pathogens [132,133]. In IBD, *NOD2* mutations are associated with decreased release of defensins [134]. NOD2-mediated autophagy is important for the generation of major histocompatibility complex (MHC)-II-restricted CD4+ T cell responses in dendritic cells, and patients with CD with high-risk *NOD2* or *ATG16L1* variants exhibit impaired MHC II antigen presentation [124].

IL-23 signaling affects both the innate and adaptive immune systems in mice and is required for colitis development in several models [135,136,137]. The dominant *IL23R* SNP protects against IBD and generates a soluble receptor antagonist of IL-23 [138]. Variants in the autophagy-associated *IRGM* gene interfere with Paneth cell morphology and function, and are associated with abnormal secretory granule development, decreased antimicrobial peptide production, and higher susceptibility to colitis in a DSS-induced model [139]. Mutations in *PTPN2* lead to defective autophagosome formation and bacterial elimination and promote T cell differentiation into Th1 and Th17 types [140,141,142]. Patients with IBD with *PTPN2* variants demonstrate increased levels of interferon (IFN)-γ, IL-17, and IL-22 in the serum and intestinal mucosa [143]. *LRRK2* is involved in the activation of dendritic cells (DCs) and production of IL-2 and TNF-α in CD [144].

## 5. Epigenetic Modifications

Recent data have provided the basis for the hypothesis that epigenetic modifications, resulting from interactions between the host and exposome, determine the phenotypic expression of IBD. For instance, the relatively high discordance rate among monozygotic twins [145] and an increased risk of developing the disease among people migrating from low- to high-incidence regions of IBD [146] constitute important epidemiological information to support the pathogenic role of epigenetic changes. Consequently, epigenetic factors have been suggested to mediate critical interactions between the exposome and genome, offering new insights into the pathogenesis of several diseases, including IBD [147].

Epigenetic changes related to the gut microbiome include modifications to DNA or histones, as well as the regulation of non-coding RNAs [148]. For example, recent studies have shown that microorganisms can bind to lysine on histones and regulate host chromatin by modifying histone proteins [149]. In turn, post-translational modifications of histones induced by microorganisms lead to changes in transcriptional gene activity [150,151]. Other studies investigating microRNAs (miRNAs) have suggested their participation in the immune response to microorganisms, resulting in the regulation of inflammatory mediators [152]. For example, miR-10a has been shown to suppress CD4+ T cell production of IL-10, favoring the induction of more severe colitis in genetically predisposed *Rag1*^−/−^ mice [153]. In addition, miR-155 has been shown to promote Th17 differentiation and upregulate Th17-related cytokines [154]. Moreover, the induction of miR-155 and miR-146 family members has been implicated in the regulation of inflammatory responses triggered by microorganisms [155,156].

Dietary components also promote epigenetic modifications either directly or through the action of the gut microbiome, as some metabolites may modulate gene expression, chromatin remodeling, and DNA methylation. For example, polyphenols in green tea or soybeans, such as epigallocatechin-3-gallate and genistein, have been shown to inhibit DNA methyltransferase activity. Additionally, the gut microbiome generates a variety of SCFAs, such as acetate, butyrate, and propionate, which are essential for epithelial cell homeostasis but can also epigenetically regulate the immune response [157]. Bacteria from *Clostridium*, *Eubacterium*, and *Butyrivibrio* genera can synthesize butyrate, which inhibits histone deacetylases, from non-digestible fibers in the gut lumen [158]. In addition to being a nutrient for epithelial cells, SCFAs can also induce intracellular signaling pathways through the activation of G-protein-coupled receptors, regulating cell metabolism, inflammation, and oxidative stress [159,160]. Furthermore, the gut microbiome also contributes to the absorption and secretion of minerals, such as iodine, zinc, selenium, cobalt, and other cofactors that participate in epigenetic processes. Additionally, other key metabolites of the gut microbiota, including S-adenosylmethionine, acetyl-coenzyme A, nicotinamide adenine dinucleotide, α-ketoglutarate, and adenosine triphosphate, serve as essential cofactors for epigenetic enzymes that regulate DNA methylation and histone modifications [161,162].

## 6. Inappropriate Immune Response

The epithelium and its specialized cell types act as a barrier, separating the microbiota in the lumen from the immune cells in the lamina propria. In this reciprocal relationship, the microbiota also produces the metabolites necessary for epithelial cells, such as SCFAs and bacteriocins [163]. Immune cells in the lamina propria constitute the mucosa-associated lymphoid tissue that responds to microbiota stimuli, along with the epithelium. Some innate immune cells such as DCs, macrophages, natural killer cells, and innate lymphoid cells (ILCs) sample lumen antigens and induce a tolerogenic immune response. Under homeostatic conditions, this immune surveillance does not initiate a proinflammatory response; in contrast, it induces Treg cells. These immune cells capture antigens and migrate to lymphoid tissues to activate lymphocytes and link innate and adaptive responses [163,164]. The interaction between the immune system and microbiota, especially segmented filamentous bacteria, symbiotically aids the maturation of the immune system [164,165].

The dysregulated immune response observed in IBD is thought to result from crosstalk among genetic susceptibility, environmental factors, and gut microbiota [166,167]. Dysbiosis changes the composition of the gut microbiota, resulting in the loss of commensal bacteria and growth of pathogenic microorganisms [17,168]. In IBD, dysbiosis is characterized by a decrease in alpha diversity, with a decrease in abundance of Bacteroidetes and Firmicutes and an increase in that of *Gamma-proteobacteria*, especially AIEC [114,169]. Moreover, dysbiosis in IBD leads to a shift towards a proinflammatory environment with activated immune cells. In this context, cells increase the expression levels of pattern recognition receptors (PRRs) and the production of proinflammatory mediators in response to pathogens [170]. PRRs recognize microbe-, pathogen-, and/or danger-associated molecular patterns (MAMPs, PAMPs, and DAMPs, respectively, e.g., ATP and high mobility group box 1 protein (HMGB1)) released by cells during inflammation [171]. Examples of MAMPs include LPS (a TLR-4 ligand) and flagellin (a TLR-5 ligand) from bacteria, β-1,3-glucans (a dectin-1, C-type lectin receptor (CLR) ligand) from fungi, and viral nucleic acid molecules [172]. TLRs and CLRs are distributed on the surface of immune cells, epithelial cells, and other cell types, whereas NOD-like receptors (NLRs) and RIG-I-like receptors are present in the cytoplasm [173,174].

Tissue-resident macrophages comprise a large population of macrophages in the gut and control either tolerance or defense against microorganisms. Therefore, macrophages exhibit specialized gene expression related to their localization within the intestinal mucosa [175,176]. ILCs represent a group of innate immune cells, mostly localized at mucosal sites, with important participation in immune-mediated diseases such as IBD. In addition to increasing in density in the inflamed mucosa, ILC-1-producing IFN-γ cells characteristically accumulate in CD [177]. Because of their proximity to the gut microbiome, mucosal ILCs are thought to participate in a dichotomous regulatory mechanism, in which ILCs interfere with the microbial composition of the gut, and the gut resident microbes shape the plasticity and physiological functions of ILCs [178].

Inflammasomes have recently been regarded as central and specifically attractive in IBD immunopathogenesis because of their participation in complex crosstalk between the host mucosal immune system and environment, particularly the microbiota. Inflammasomes are multiprotein platforms formed in the cytoplasm that cleave and activate caspase-1, leading to the production of inflammatory cytokines, including IL-1b and IL-18. Inflammasomes comprise intracellular sensors formed by NLR proteins NLRP1, NLRP3, NLRC4, NLRP6, and NAIP5, or by the DNA-sensing complex AIM2, and can be activated by extracellular and intracellular pathogens in the presence of DAMPs [179]. Inflammasomes participate in responses against several bacteria such as *L. monocytogenes*, *M. tuberculosis*, and *Fusobacterium* [180]. In patients with CD, increased NLRP3 and AIM2 activity has been reported, and an SNP in *NLRP3* is associated with CD susceptibility [181,182]. In contrast, NLRP3 inflammasome activation and production of IL-1b and IL-18 appear to be protective in experimental IBD [183].

Together, through these mechanisms, innate immune cells sense PAMPs and DAMPs, induce an inflammatory response, and shape adaptive immunity, promoting lymphoid tissue expansion and T- (Th1, Th2, Th9, Th17, and Treg cells) and B-cell responses [184]. Additional direct and indirect participation of gut microbiota through the metabolism of dietary vitamins and SCFAs, such as butyrate, also influences the immune response, promoting Treg differentiation and tolerance [185,186]. For example, appropriate Th17 and Th1 responses are important for the clearance of *Citrobacter rodentium* and non-typhoidal *Salmonella enterica* infections [187,188]. Although, under normal conditions, the gut constitutes a microenvironment controlled by balanced T-cell responses, prolonged dysbiosis favors an inappropriate persistent proinflammatory response [189].

## 7. Microbial-Based Therapies

Antibiotics have long been considered in the treatment of IBD because of the prevalence of microbial abnormalities and the presence of known pathogens. Although the use of antibiotics has been clearly supported for the treatment of infectious complications related to IBD, several pieces of evidence have failed to find consistent beneficial associations between antibiotic treatment and IBD remission [190]. Selby et al. did not find beneficial outcomes for CD with treatment with clarithromycin, rifabutin, and clofazimine aimed at eradicating *M. avium* subspecies *paratuberculosis* in a two-year randomized clinical trial [191]. Nevertheless, some data support the clinical application of antibiotics such as ciprofloxacin, with or without metronidazole, for treating active fistulizing perianal CD [21,192].

Other methods of bacterial manipulation have provided additional evidence supporting the role of the microbiota in the pathogenesis of IBD. One method of microbiota manipulation in IBD is the introduction of dietary probiotics to control the growth of pathological components and/or switch the global composition towards a healthier one. *E. coli* Nissle 1917, a nonpathogenic strain clinically used as a probiotic, has been shown to be effective in inducing the remission of patients with UC. In addition, *E. coli* Nissle 1917 has been associated with maintenance of remission in patients with UC for at least one year [193,194]. Similarly, the probiotic VSL#3, a set of eight bacterial strains (*Bifidobacterium breve*, *B. longum*, *B. infantis, Lactobacillus acidophilus, L. plantarum, L. paracasei, L. bulgaricus*, and *Streptococcus thermophilus*) has significantly reduced scores of disease severity and induced remission in patients with UC compared to a placebo [37,195,196]. Other probiotics, such as *Lactobacillus* GG, have been shown to be effective when associated with IBD oral therapy, such as mesalamine [38,197]. Nevertheless, so far, data regarding the effectiveness of probiotics for treating patients with CD have failed to reach substantial association with the induction of remission.

Although currently available probiotics potentially modulate dysbiosis in IBD, their effects are transient and limited in most IBD subsets. In fact, most existing probiotics encounter colonization resistance in the host intestine and are present only for a limited period, even after long-term administration. Therefore, new alternatives have been investigated, including the use of genetically modified organisms with recombinant bacteria as vectors to deliver therapeutic molecules at target sites in the gut. For example, modified strains of *Lactobacillus casei* BL23 [198] and *Streptococcus thermophilus* CRL 807 [199] were engineered to produce superoxide dismutase, which has anti-inflammatory properties. *Lactococcus lactis*-secreting IL-10 [200], elafin (a human protease inhibitor) [201], and IL-27 (an immunosuppressive cytokine) [202] reportedly have anti-inflammatory effects in colitis models and may therefore represent potential candidates for future clinical trials.

Another method of microbiota manipulation of increasing interest for potential therapeutic applications in various diseases is fecal microbiota transplantation (FMT). Recently, randomized clinical trials have assessed the benefits of the use of FMT in the treatment of patients with IBD. Moayyedi et al., for example, found that patients with recently diagnosed UC could be induced to remission after treatment with FMT [203]. Data from another study showed that for patients in remission, treatment with FMT was able to maintain clinical remission in 87.1% of patients compared to 66.7% receiving a placebo. These results indicate that the long-term beneficial effect of FMT in patients with UC in clinical remission could help sustain endoscopic, histological, and clinical remission [204]. A recent meta-analysis showed that FMT was effective in promoting clinical remission (OR = 3.47, 95% CI = 1.93–6.25) and clinical response (OR = 2.48, 95% CI = 1.18–5.21) to patients with active UC when compared to placebo [205]. Studies on the effectiveness of therapeutic microbiota manipulation in IBD are still in progress, and the results are expected to further understanding and guide the potential application in clinical practice.

## 8. Complex Genetic and Molecular Network

As previously mentioned, by using modern sequencing methodologies, several studies have compared and described the microbiota composition of healthy individuals versus patients with IBD [17,50,168]. For example, recent data revealed a loss of microbial diversity in patients with IBD, with a clear separation between CD and healthy patients, and a more heterogeneous profile in patients with UC [168]. Antibiotic resistance gene levels were increased in IBD, and their abundance was positively correlated with *Escherichia* and *Bacteroides* bacteria [206]. Furthermore, higher than normal levels of hydrogen sulfide generated by gut microbiota have been strongly associated with IBD pathogenesis and indicate increased prevalence of sulfate-reducing bacteria, such as *Deltaproteobacteria*, *Desulfotomaculum*, *Desulfosporosinus*, *Thermodesulfobacterium*, and *Thermodesulfovibrio* genera [207].

In CD, metagenomic and metaproteomic studies have characterized a decrease in levels of butanoate and propanoate metabolism genes, butyrate, and other SCFAs, in agreement with the decrease in abundance of SCFA-producing Firmicutes bacteria seen in taxonomic profiling studies [208,209]. Several functional changes in the microbiome of IBD have been identified, including an increase in the activities of pathobionts, alterations in the synthesis of amino acids, neurotransmitters, and vitamins, regulation of mineral absorption, degradation of complex carbohydrates, and effects on pathways related to SCFAs, cysteine, and L-arginine synthesis [168,209,210]. In a metagenomic study with a pediatric CD cohort undergoing anti-TNF-α therapy, greater microbiota changes were correlated with higher levels of fungal and human DNA and variations in microbial genes. Examples of these variations include a decrease in selenocompound metabolic pathway activity and an increase in levels of microbial genes encoding glycerophospholipid metabolism, aminobenzoate degradation, sulfur relay systems, and glutathione metabolism [211]. Together, these data indicate that further metabolomic studies could help differentiate, diagnose, and better characterize disease activity [212].

The measurement of RNA transcripts in tissues from patients with IBD may predict the pathways that are activated and involved in the disease. Using deep RNA sequencing, studies of the transcripts have identified molecular subtypes of CD [213,214,215]. A remarkable report on this topic is the Pediatric RISK Stratification Study, which showed that these molecular signatures may predict disease behavior [216]. However, as RNA does not necessarily represent the proteins produced in cells, there are limitations to this approach. For example, a study on the feasibility of metatranscriptomics for fecal samples observed that transcriptional profiles differed more between individuals than metagenomic functional profiles [168]. Metatranscriptomic data also revealed some species-specific biases in the transcriptional activity of gut bacteria, especially with IBD-specific microbial populations, such as *F. prausnitzii* [210].

Metabolomics research, including analysis of plasma, serum, urine, stool, and intestinal biopsies, has provided data allowing for differentiation between healthy controls and patients with IBD [217]. In the stool of patients with IBD, a loss of metabolites has been observed in concordance with the loss of microbial diversity [168,218]. There were lower levels of secondary bile acids, sphingolipids, short- and medium-chain fatty acids, and vitamins, whereas primary bile acids, amino acids, polyamines, arachidonate, and acylcarnitines were present in higher levels compared to the controls [219]. In IBD, farnesoid X receptor (FXR) activation, triggered by bile salts, led to the downregulation of proinflammatory cytokines, and in CD, intestinal biopsies showed lower expression levels of FXR [220,221]. Another compound commonly associated with IBD is tryptophan, an essential aromatic amino acid obtained from the diet and a precursor of numerous molecules, such as serotonin, melatonin, nicotinamide, and vitamin B3, as well as other intermediates. Common sources of tryptophan are dairy foods, poultry, fish, and oats, and tryptophan is metabolized by both the host and gut microbiota. Microbiota metabolism leads to indole metabolites that can activate aryl hydrocarbon receptors (AhR) and participate in the onset of IBD [222,223]. Few bacteria produce AhR agonists, such as *Peptostreptococcus russellii* and members of *Lactobacillus*, whereas indole-propionic acid (IPA) production has been best characterized in *Clostridum sporogenes* [224]. Indole induces the release of glucagon-like peptide-1 and its derivatives, indoleacetic acid, indole-3-acetaldehyde, indole-3-aldehyde, indoleacrylic acid, and indole-3-propionic acid. IPA via AhR affects T-cell immunity and exerts anti-inflammatory effects in the gut [225]. AhR expression levels are reduced in patients with CD, whereas tryptophan deficiency promotes more severe colitis in mice [226,227].

Proteomic studies have also been conducted to explore innate and adaptive immune mechanisms in IBD. For example, compared to those in the controls, in patients with UC, 46 proteins, excluding neutrophils and their extracellular trap proteins, were more abundant in the colon tissue [228,229]. In intestinal biopsies of patients with CD, the proteomes of human Th1 and Th1/Th17 clones were studied, and 334 proteins were found to be differentially expressed. Cytotoxic proteins, such as granzyme B and perforin, were more abundant in Th1 cells than in Th17 cells, but only in a subgroup of Th1 cell clones from patients with CD [230]. Regarding regulatory T cells (CD4+ Foxp3+), a proteomics study identified a novel protein, THEMIS, which is important for the suppressive function of Treg cells [231]. In agreement with proteomic studies, the lipidome and immune responses have also been investigated in IBD. For instance, the inflamed mucosa of patients with UC showed increased levels of seven eicosanoids (prostaglandin (PG) E2, PGD2, thromboxane B2, 5-hydroxyeicosatetraenoic acid (HETE), 11-HETE, 12-HETE, and 15-HETE) [232]. Macrophages from patients with CD challenged with heat-inactivated *E. coli* presented lower levels of newly synthesized phosphatidylinositol [233]. Lipidomic analysis of the phosphatidylcholine lipidome profile of rectal mucus obtained from patients with UC showed lower levels of phosphatidylcholine compared to patients with CD and controls. Interestingly, supplementation with delayed-release phosphatidylcholine was clinically effective [234].

Although knowledge of the mechanisms underlying IBD continues to expand, novel data stemming from individual pathogenic constituents are usually not integrated, leading to only limited data being utilized for achieving relevant progress in the field [64]. Regarding the microbiome, complexity becomes even more evident as modern evolving technologies provide an exponential increase in novel information with an overwhelming accumulation of data. Hence, it is currently believed that a better understanding of the pathogenesis of complex diseases such as IBD will depend on the comprehensive integration of knowledge from different “–omes,” including the microbiome, exposome, and genome.

## 9. Conclusions

In the last two decades, it has become increasingly evident that the microbiome, immune system, genome, and exposome are comprised of highly complex, dynamic, and mutually interactive systems. Nevertheless, the traditional approach for evaluating the individual components that presumably participate in the pathogenesis of IBD, including the microbiota, has not been sufficient to determine the interconnecting pathways underlying the multiple biological systems involved in the disease development. Even using the best of the currently available methods, including clinical, laboratory, endoscopic, histological, and imaging parameters, we still only have a narrow understanding of the intricate mechanisms responsible for chronic inflammation and the peculiar dynamics and specificities affecting each patient with IBD. Consequently, current treatments continue to be mostly empirical and have limited efficacy.

Regarding the microbial component, although causality remains to be clearly established, evidence indicating an association with IBD pathogenesis is rapidly accumulating. However, a better understanding of the probable microbial basis of IBD depends on more complete, deep, and unbiased investigations at multiple and simultaneous levels, including microbial strains and genomic and functional features, ideally allowing the construction of full transcriptomic and metabolomic profiles. High-throughput technologies capable of analyzing innumerable parameters of the microbiome in conjunction with other system variables have been developed in recent years. Hopefully, more integrative analysis will enable data assembly in a comprehensive fashion to build an IBD network and translate information into useful biological insights with direct influence on specific therapeutic targets, clinical decisions, and disease outcomes, which will preferably be individualized.

## Figures and Tables

**Figure 1 ijms-23-03464-f001:**
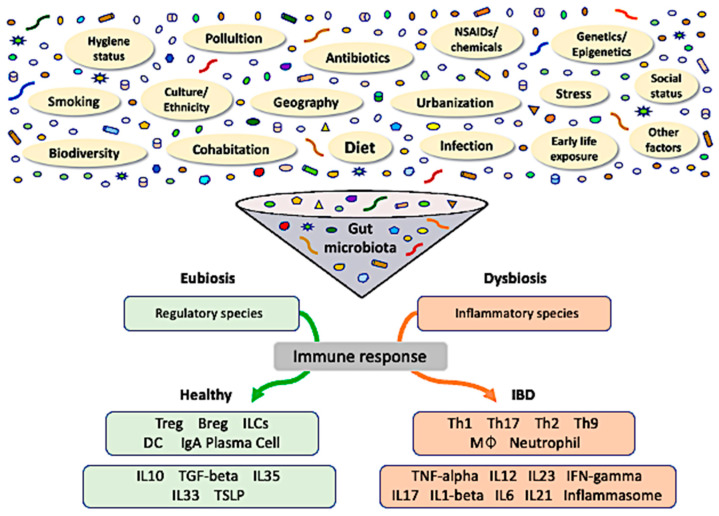
The role of gut dysbiosis in the pathogenesis of inflammatory bowel disease. Gut microbiota reflect an interaction of host genetics with dynamic exposure to innumerable stimuli from the exposome. Crosstalk amongst these factors results in long-standing consequences to the gut microbiota and epigenetic modifications in a multidirectional fashion, potentially affecting susceptibility to diseases. The prevalence of either regulatory (eubiosis) or inflammatory (dysbiosis) species within the gut microbial community determines the respective predominant immune response. Treg, regulatory T-cell; Breg, regulatory B-cell; ILC, innate lymphoid cell; IgA, immunoglobulin A; MØ, macrophage; TSLP, thymic stromal lymphopoietin.

**Table 1 ijms-23-03464-t001:** Association between the gut microbiome and inflammatory bowel disease.

Microbiome Components	Presence in IBD	Possible Mechanisms	Evidence	References
Firmicutes				
*Faecalibacterium prausnitzii*	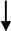	It is a highly active metabolic commensal bacterium involved in the production of butyrate. This metabolite plays a major role in gut physiology and has beneficial effects, including protection against pathogen invasion, modulation of the immune system, and promotion of anti-inflammatory activity	Presence of *F. prausnitzii* may serve as a biomarker of intestinal health in adults. Low levels of this bacteria could be predictive for CD. Its deficiency was shown in colonic CD. Low *F. prausnitzii* levels in patients with IBD undergoing surgery is associated with a higher risk of post-operative recurrence	[25,27]
*Eubacterium spp.*	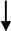	Involved in the production of SCFAs, especially butyrate. It is important in inflammation modulation and the promotion of epithelial barrier integrity	Found deficient in samples from patients with CD and UC	[24,28]
*Ruminococcus albus*	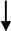	Possibly involved in SCFA metabolism and its protective and anti-inflammatory roles	Found decreased in samples from patients with CD and UC	[24,29]
*Ruminococcus gnavus*	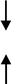	Involved in bile and amino acid biosynthesis pathways, including amino acid, energy, carbohydrate, and nucleotide metabolismLack of supporting evidence of possible mechanisms involved	Found decreased in the stool of patients with treatment-naïve new-onset CDFound increased in samples from patients with CD compared to controls	[24,30]
*Clostridioides difficile*	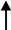	A and B toxins produced by this bacterium may activate caspase-1 and secrete mature IL-1b and IL-18 (proinflammatory cytokines) that cause damage to the epithelial barrier and intestinal cells	High prevalence of infection by *C. difficile* has been demonstrated among patients with IBD	[31,32]
*Listeria monocytogenes*	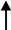	Possibly associated with invasive infection of epithelial cells	*Listeria monocytogenes* infection rates seem to be elevated in patients with IBD	[31,33]
Verrucomicrobia				
*Akkermansia muciniphila*	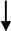	Possibly involved with the production of SCFAs, which can activate the GPR43 and thereby increase the number of Foxp3+ regulatory T cells in the colon	Decreased in stools of both CD and UC patients.Human strain ATCC BAA-835T and murine strain 139 exerted anti-inflammatory effects onDSS-induced chronic colitis in mice	[34,35,36]
Actinobacteria				
*Eggerthella lenta*	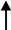	Lack of well-explored possible mechanisms	Found increased in samples from patients with CD compared to controls	[24]
*Bifidobacterium bifidum*	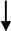	Mucin metabolism performed by *B. bifidum* could activate enhanced production of mucin, thereby increasing the mucus layer depth and strengthening the epithelial barrier function	Found decreased in samples from patients with IBD. Some studies suggest that probiotics containing this bacterium could have positive responses in the treatment of IBD	[37,38]
*Mycobacterium avium*	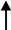	Associated with increased production of proinflammatory cytokines. Mutations in NOD2/CARD15 receptors may cause intracellular survival of the bacteria and ultimately cause infection	The abundance of this bacteria, especially the subspecies *paratuberculosis*, is higher in patients with IBD than in controls	[39,40]
Proteobacteria				
*Escherichia coli*	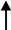	Epithelium-associated invasive *E. coli* has frequently been isolated from ileal and colonic mucosa from patients with CD and can infect and damage intestinal epithelial cell monolayers, and synthesize α-hemolysin	Found increased numbers of *E. coli*strains with virulence properties isolated from samples of patients with IBD. Several studies indicate that there is a link between the prevalence of *E. coli* and IBD relapses	[41,42]
*Haemophilus parainfluenzae*	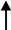	Involved in glycerol-phospholipidand lipopolysaccharide metabolism, thereby promoting inflammation	Found increased in stool samples from patients with treatment-naïve new-onset CD	[30]
*Campylobacter spp.*	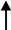	Invasive strains of this bacteria in patients with IBDcan survive in intracellular andanaerobic conditions	Increased in patients with IBD compared to controls	[43]
*Eikenella corrodens*	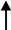	Possibly involved in lipid and polysaccharide metabolism, resulting in proinflammatory responses	Increased in patients with IBD compared to controls	[30]
Fusobacteria				
*Fusobacterium nucleatum*	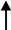	Possibly involved in proinflammatory and tumorigenic responses	Increased in patients with IBD compared to controls. It is also associated with colorectal cancer	[43]
Bacteroidetes				
*Bacteroides spp.*	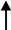	Involved in mucin metabolism, possibly playing a role in damaging the protective mucus layer	Increased in CD samples. Prominent in patients with prior surgical resection	[24,44,45,46]

Abbreviations: CD, Crohn’s disease; G protein-coupled receptor 43 (GPR43); IBD, inflammatory bowel disease; SCFAs, short-chain fatty acids; UC, ulcerative colitis; IL, interleukin; NOD2/CARD15, nucleotide-binding oligomerization domain 2/caspase recruitment domain family, member 15.

## Data Availability

Not applicable.

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
