# Peer review of "Dysbiosis in Inflammatory Bowel Disease: Pathogenic Role and Potential Therapeutic Targets"

_ijms, 2022, doi:10.3390/ijms23073464_

Round 1

Reviewer 1 Report

Santana et al. is a review aimed to discuss and summarize the basic mechanisms and potential associations between dysbiosis and IBD development, as well as exposome influences, methodological approaches to the microbiome, epigenetic changes, and recent developments in therapeutics. Gut-luminal microbiota reflect an interaction of host genetics with dynamic exposure to innumerable stimuli from the exposome. Crosstalk amongst these factors results in long-standing consequences to the gut-luminal microbiota and epigenetic modifications in a multidirectional fashion, potentially affecting susceptibility to diseases. The prevalence of either regulatory (eubiosis) or inflammatory (dysbiosis) species within the gut-luminal microbial community determines the respective predominant immune response (the antibody-antigen reaction against the mucosa resistance of a susceptible individual). I truly enjoyed reviewing this manuscript which is well summarized supported by relevant citations. The fundamental question remains obscure as to why the immune system responds aggressively to harmless, ever-present bacteria, releasing complex mixes of cytokines, chemokines, growth factors and other substances that cause inflammation [PMID25429322].

This article is publishable in its current form.

Author Response

R: We thank this reviewer for his/her comments, for supporting our work, and for finding our manuscript interesting.

Reviewer 2 Report

  • Use new name Clostridioides difficile

  • Is the function of acetate positive or negative for our bowel?

  • Add Akkermansia in Table 1

  • “Moreover, microbiome comparison between patients with CD with and without antibiotic exposure indicated that antibiotic use amplified the microbial dysbiosis associated with CD.”

Is this an age-dependent factor?

  • What about Archaea role?

  • Cite the role of IBD drug in microbiome (see “Adalimumab Therapy Improves Intestinal Dysbiosis in Crohn's Disease. J Clin Med. 2019 Oct 9;8(10):1646. doi: 10.3390/jcm8101646.”)

  • “This may explain why their reduction in a typical Western-style diet (animal-based, high-calorie, high-fat, and low-dietary fiber) has been associated with the risk of IBD”

Why did you use the term reduction?

  • “Escherichia coli Nissle 1917, Lactobacillus GG, Bifidobacterium strains, and Saccharomyces boulardii are relatively effective in the treatment of IBD, at least temporarily.”

This sentence is too generic. IBD or UC? In what specific setting? For example, Escherichia coli Nissle 1917 has data about maintain remission in UC

  • “Several controlled clinical trials are ongoing worldwide, but currently, only uncontrolled observations, such as case reports and open-label studies, are available concerning FMT for treating IBD.”

This is false. At least three RCT of FMT in UC are available

Author Response

Reviewer#2

Comments and Suggestions for Authors

  • Use new name Clostridioides difficile

R: This reviewer is correct, and we agree with his/her suggestion. The correction was made in Table 1.

  • Is the function of acetate positive or negative for our bowel?

R: The current knowledge supports the overall concept that short-chain fatty acids produced by the intestinal microbiota through the digestion of fibers have fundamentally an anti-inflammatory action.

Acetate from our diet or produced by bacteria in our microbiota is a cell energy source involved in the generation of cholesterol and acetyl-CoA with overall positive effects in our gut. Acetate is known to have important regulatory roles in body weight control and insulin sensitivity through effects on lipid metabolism and glucose homeostasis, on epithelial barrier function, and immune regulation, acting by binding to the G-protein coupled receptors (GPR), GPR43 (FFAR2), and GPR41 (FFAR3). Acetate has antiinflammatory properties (Tedelind et al., 2007) and is associated with bacteria-specific IgA production and regulatory T cell and B cell production (Takeuchi et al., 2021; Daïen et l., 2021). Also, some protective bacteria such as Faecalibacterium prausnitzii and Roseburia intestinalis are acetate consumers and use it to produce butyrate, with clear beneficial effects to our bowel (Duncan et al., 2004). However, currently, the role of acetate in human IBD is not fully understood. While acetate attenuated colitis in mice (Laffin et al., 2019), some synergy has been observed between H2S and acetate produced by sulfate-reducing bacteria (Kushkevych et al., 2020), suggesting that in inflammatory conditions, dysbiosis and excess of acetate, may have harmful effects. Moreover, it is also likely that differential local concentrations may influence the biological action of acetate in the gut.

REFERENCES

  • Tedelind S, Westberg F, Kjerrulf M, Vidal A. Anti-inflammatory properties of the short-chain fatty acids acetate and propionate: a study with relevance to inflammatory bowel disease. World J Gastroenterol. 2007 May 28;13(20):2826-32. doi: 10.3748/wjg.v13.i20.2826. PMID: 17569118; PMCID: PMC4395634.
  • Takeuchi T, Miyauchi E, Kanaya T, Kato T, Nakanishi Y, Watanabe T, Kitami T, Taida T, Sasaki T, Negishi H, Shimamoto S, Matsuyama A, Kimura I, Williams IR, Ohara O, Ohno H. Acetate differentially regulates IgA reactivity to commensal bacteria. Nature. 2021 Jul;595(7868):560-564. doi: 10.1038/s41586-021-03727-5. Epub 2021 Jul 14. PMID: 34262176.
  • Daïen CI, Tan J, Audo R, Mielle J, Quek LE, Krycer JR, Angelatos A, Duraes M, Pinget G, Ni D, Robert R, Alam MJ, Amian MCB, Sierro F, Parmar A, Perkins G, Hoque S, Gosby AK, Simpson SJ, Ribeiro RV, Mackay CR, Macia L. Gut-derived acetate promotes B10 cells with antiinflammatory effects. JCI Insight. 2021 Apr 8;6(7):e144156. doi: 10.1172/jci.insight.144156. PMID: 33729999; PMCID: PMC8119207.
  • Duncan SH, Holtrop G, Lobley GE, Calder AG, Stewart CS, Flint HJ. Contribution of acetate to butyrate formation by human faecal bacteria. Br J Nutr. 2004 Jun;91(6):915-23. doi: 10.1079/BJN20041150. PMID: 15182395.
  • Laffin M, Fedorak R, Zalasky A, Park H, Gill A, Agrawal A, Keshteli A, Hotte N, Madsen KL. A high-sugar diet rapidly enhances susceptibility to colitis via depletion of luminal short-chain fatty acids in mice. Sci Rep. 2019 Aug 23;9(1):12294. doi: 10.1038/s41598-019-48749-2. PMID: 31444382; PMCID: PMC6707253.
  • Kushkevych I, Dordević D, Vítězová M. Possible synergy effect of hydrogen sulfide and acetate produced by sulfate-reducing bacteria on inflammatory bowel disease development. J Adv Res. 2020 Mar 24;27:71-78. doi: 10.1016/j.jare.2020.03.007. PMID: 33318867; PMCID: PMC7728581.

  • Add Akkermansia in Table 1

R: We understand this reviewer’s concern, and we amended the respective Table to accommodate the change suggested. As a result, we added two more lines to Table 1 and 3 new respective references.

  • Lo Presti, A. et al, (2019). Exploring the genetic diversity of the 16S rRNA gene of Akkermansia muciniphila in IBD and IBS. Future Microbiology, 14, 1497–1509.
  • Zhai, R., et al. (2019). Strain-specific anti-inflammatory properties of two Akkermansia muciniphila strains on chronic colitis in mice. Frontiers in Cellular and Infection Microbiology, 9, 1–12.
  • Zhang, T., et al. (2021). The potential of Akkermansia muciniphila in inflammatory bowel disease. Applied Microbiology and Biotechnology, 105, 5785–5794.

  • “Moreover, microbiome comparison between patients with CD with and without antibiotic exposure indicated that antibiotic use amplified the microbial dysbiosis associated with CD.”

Is this an age-dependent factor?

R: We understand this reviewer’s concern, and we attempted to contemplate the issue by amending the paragraph. We corrected the sentence to clarify the meaning and included additional information at the end of the paragraph, with a new reference (Zhuang et al, 2021).

Correction: “Moreover, microbiome comparison between patients with CD with and without antibiotic exposure indicated that antibiotic use amplified the microbial dysbiosis associated with CD in a new-onset pediatric cohort.”

New additional sentence (end of paragraph): “Although changes in the gut microbiota profile in new-onset and treatment-naive pediatric patients with IBD was further corroborated in a recent systematic review, no clear conclusion can be drawn at the moment, due to inconsistent results and heterogeneous methodologies (Zhuang et al, 2021).”

REFERENCES

  • Zhuang X, Liu C, Zhan S, Tian Z, Li N, Mao R, Zeng Z, Chen M. Gut Microbiota Profile in Pediatric Patients With Inflammatory Bowel Disease: A Systematic Review. Front Pediatr. 2021 Feb 2;9:626232. doi: 10.3389/fped.2021.626232. PMID: 33604319; PMCID: PMC7884334.

  • What about Archaea role?

R: We again understand this reviewer’s concern. Therefore, we attempted to summarize current knowledge on the role of Archaea updating and amending the manuscript, including 4 new references (please refer to the new paragraph, the first one on page 7).

The majority of the archaea members in the human gut are methane-producing organisms (methanogens), which use H2 to produce methane under anaerobic conditions. Methanogens establish a syntrophic relationship with bacteria, consuming H2 and improving bacterial fermentation efficiency. Some archaea members which have been described in the human stools are Methanobrevibacter smithii, Methanosphaera stadtmanae, and Methanomassiliicoccus luminyensis and in the oral mucosa, Methanobrevibacter oralis (Miller et al., 1982; Miller et al., 1985; Nkamga et al., 2017; Matijašić et al., 2020). Molecular tools allowed the characterization of not only methanogens of the orders Methanosarcinales, Methanobacteriales, Methanococcales, Methanomicrobiales, and Methanopyrales (Gaci et al., 2014), but also non-methanogens of the orders Desulfurococcales, Sulfolobales, Thermoproteales, Nitrososphaerales, and Halobacteriales in the human microbiome (Gaci et al., 2014). However, only Haloferax massiliensis and Haloferax assiliense were successfully isolated from the human gut (Khelaifia et al., 2016; Khelaifia et al., 2018). M. luminyensis has been proposed as an “archaebiotic” because it can degrade trimethylamine (TMA) and reduce trimethylamine-N-oxide (TMAO) plasma levels, thus helping in cardiovascular and chronic kidney diseases (Brugere et al., 2014). In the gut, M. smithii abundance is related to constipation in irritable bowel disease (Ghosal et al., 2016), whereas in IBD, methanogens seem to be reduced, especially M. smithii (Scanlan & Marchesi, 2008). The immunogenic M. stadtmanae, however, is three-fold increased in patients with IBD compared to healthy controls (Blais et al., 2014). These changes may be related to bacterial dysbiosis, which favors methylotrophic archaeal species such as M. stadtmanae (Burman et al., 2016) or to the “syntrophic imbalance hypothesis”, in which archaeal overgrowth would remove SCFA, especially butyric acid, from the biofilms, leading bacteria to become invasive and enter intestinal epithelial tissues (Matijašić et al., 2020).

REFERENCES

  • Miller TL, Wolin MJ, Conway de Macario E, Macario AJ. Isolation of Methanobrevibacter smithii from human feces. Appl Environ Microbiol. 1982 Jan;43(1):227-32. doi: 10.1128/aem.43.1.227-232.1982. PMID: 6798932; PMCID: PMC241804. 
  • Miller TL, Wolin MJ. Methanosphaera stadtmaniae gen. nov., sp. nov.: a species that forms methane by reducing methanol with hydrogen. Arch Microbiol. 1985 Mar;141(2):116-22. doi: 10.1007/BF00423270. PMID: 3994486.
  • Matijašić M, Meštrović T, Paljetak HČ, Perić M, Barešić A, Verbanac D. Gut Microbiota beyond Bacteria-Mycobiome, Virome, Archaeome, and Eukaryotic Parasites in IBD. Int J Mol Sci. 2020 Apr 11;21(8):2668. doi: 10.3390/ijms21082668. PMID: 32290414; PMCID: PMC7215374. 
  • Nkamga,V.D.; Henrissat, B.; Drancourt, M. Archaea: Essential inhabitants of the human digestive microbiota. Hum. Microbiome, J. 2017, 3, 1–8. 
  • Gaci N, Borrel G, Tottey W, O'Toole PW, Brugère JF. Archaea and the human gut: new beginning of an old story. World J Gastroenterol. 2014 Nov 21;20(43):16062-78. doi: 10.3748/wjg.v20.i43.16062. PMID: 25473158; PMCID: PMC4239492.
  • Khelaifia, S.; Caputo, A.; Andrieu, C.; Cadoret, F.; Armstrong, N.; Michelle, C.; Lagier, J.C.; Djossou, F.; Fournier, P.E.; Raoult, D. Genome sequence and description of Haloferax massiliense sp. nov., a new halophilic archaeon isolated from the human gut. Extremophiles 2018, 22, 485–498. [CrossRef] [PubMed] 
  • Khelaifia S, Raoult D. Haloferax massiliensis sp. nov., the first human-associated halophilic archaea. New Microbes New Infect. 2016 May 14;12:96-8. doi: 10.1016/j.nmni.2016.05.007. PMID: 27408734; PMCID: PMC4919280.
  • Brugère JF, Borrel G, Gaci N, Tottey W, O'Toole PW, Malpuech-Brugère C. Archaebiotics: proposed therapeutic use of archaea to prevent trimethylaminuria and cardiovascular disease. Gut Microbes. 2014 Jan-Feb;5(1):5-10. doi: 10.4161/gmic.26749. Epub 2013 Oct 31. PMID: 24247281; PMCID: PMC4049937.
  • Ghosal D, Ghosh S, Dutta TK, Ahn Y. Current State of Knowledge in Microbial Degradation of Polycyclic Aromatic Hydrocarbons (PAHs): A Review. Front Microbiol. 2016 Aug 31;7:1369. doi: 10.3389/fmicb.2016.01369. Erratum in: Front Microbiol. 2016 Nov 15;7:1837. PMID: 27630626; PMCID: PMC5006600.
  • Scanlan PD, Marchesi JR. Micro-eukaryotic diversity of the human distal gut microbiota: qualitative assessment using culture-dependent and -independent analysis of faeces. ISME J. 2008 Dec;2(12):1183-93. doi: 10.1038/ismej.2008.76. Epub 2008 Jul 31. PMID: 18670396.
  • Blais Lecours P, Marsolais D, Cormier Y, Berberi M, Haché C, Bourdages R, Duchaine C. Increased prevalence of Methanosphaera stadtmanae in inflammatory bowel diseases. PLoS One. 2014 Feb 3;9(2):e87734. doi: 10.1371/journal.pone.0087734. PMID: 24498365; PMCID: PMC3912014.
  • Burman S, Hoedt EC, Pottenger S, Mohd-Najman NS, Ó Cuív P, Morrison M. An (Anti)-Inflammatory Microbiota: Defining the Role in Inflammatory Bowel Disease? Dig Dis. 2016;34(1-2):64-71. doi: 10.1159/000443759. Epub 2016 Mar 16. PMID: 26982568.

  • Cite the role of IBD drug in microbiome (see “Adalimumab Therapy Improves Intestinal Dysbiosis in Crohn's Disease. J Clin Med. 2019 Oct 9;8(10):1646. doi: 10.3390/jcm8101646.”)

R: We thank this reviewer for his/her suggestion and we included new sentences on page 6, the end of the penultimate paragraph, lines 182-190).

The investigation of whether the response to treatment with biologic agents could be associated with alterations in the composition of the intestinal microbiota was also performed in a prospective study involving patients with CD. The therapeutic intervention based on adalimumab was associated with the restoration of a eubiotic environment after six months of treatment. Particularly, in the cohort of patients with CD, those receiving adalimumab displayed a reduction of Proteobacteria and an increase of Lachnospiraceae. These results characteristically predominated among those patients who achieved therapeutic success, suggesting that dysbiosis could be directly involved with the response to treatment (Ribaldone et al., 2019).

REFERENCE

  • Ribaldone, D. G., Caviglia, G. P., Abdulle, A., Pellicano, R., Ditto, M. C., Morino, M., Fusaro, E., Saracco, G. M., Bugianesi, E., & Astegiano, M. (2019). Adalimumab therapy improves intestinal dysbiosis in Crohn’s disease. Journal of Clinical Medicine, 8(10), 4–13. https://doi.org/10.3390/jcm8101646

  • “This may explain why their reduction in a typical Western-style diet (animal-based, high-calorie, high-fat, and low-dietary fiber) has been associated with the risk of IBD”

Why did you use the term reduction?

R: This reviewer is correct. We apologize for not being clear enough, and we agree that the phrase sounds dubious, or at least incomplete. We made amendments to clarify the meaning of the sentence (Page 9, second paragraph, line 340).

  • “Escherichia coli Nissle 1917, Lactobacillus GG, Bifidobacterium strains, and Saccharomyces boulardii are relatively effective in the treatment of IBD, at least temporarily.”

This sentence is too generic. IBD or UC? In what specific setting? For example, Escherichia coli Nissle 1917 has data about maintain remission in UC

R: We agree with this comment, and we amended the sentence to be more specific, including new references (page 13, third paragraph, lines 547-557).

  1. coli Nissle 1917, a nonpathogenic strain clinically used as a probiotic, has been shown to be effective in inducing remission of patients with UC. In addition, E. coli Nissle 1917 has been associated with maintenance of remission in patients with UC for at least one year (Matthes et al., 2010; Fedorak, 2010). Similarly, the probiotic VSL#3, a set of 8 bacterial strains (Bifidobacterium breve, B. longum, B. infantis, Lactobacillus acidophilus, L. plantarum, L. paracasei, L. bulgaricus, and Streptococcus thermophilus), has significantly reduced scores of disease severity and induced remission in patients with UC compared to placebo (Bibiloni et al., 2005). Other probiotics, such as Lactobacillus GG have been shown to be effective when associated with IBD oral therapy, such as mesalamine (Turroni et al., 2019; Wasilewski et al., 2015). Data regarding the effectiveness of probiotics for treating patients with CD, so far, have failed to reach substantial association with induction of remission.

REFERENCES

  • Fedorak, R. N. (2010). Probiotics in the Management of Ulcerative Colitis. Gastroenterology & Hepatology, 6(11), 688–690.
  • Matthes, H., Krummenerl, T., Giensch, M., Wolff, C., & Schulze, J. (2010). Clinical trial: Probiotic treatment of acute distal ulcerative colitis with rectally administered Escherichia coli Nissle 1917 (EcN). BMC Complementary and Alternative Medicine, 10. https://doi.org/10.1186/1472-6882-10-13
  • Bibiloni, R., Ph, D., Fedorak, R. N., Tannock, G. W., Ph, D., Madsen, K. L., Ph, D., Gionchetti, P., Campieri, M., Simone, C. De, Ph, D., & Sartor, R. B. (2005). VSL # 3 Probiotic-Mixture Induces Remission in Patients with Active Ulcerative Colitis. 1539–1546. https://doi.org/10.1111/j.1572-0241.2005.41794.x
  • Turroni, F., Duranti, S., Milani, C., Lugli, G. A., van Sinderen, D., & Ventura, M. (2019). Bifidobacterium bifidum: A key member of the early human gut microbiota. Microorganisms, 7(11), 1–13. https://doi.org/10.3390/microorganisms7110544
  • Wasilewski, A., Zielińska, M., Storr, M., & Fichna, J. (2015). Beneficial effects of probiotics, prebiotics, synbiotics, and psychobiotics in inflammatory bowel disease. Inflammatory Bowel Diseases, 21(7), 1674–1682. https://doi.org/10.1097/MIB.0000000000000364

  • “Several controlled clinical trials are ongoing worldwide, but currently, only uncontrolled observations, such as case reports and open-label studies, are available concerning FMT for treating IBD.”

This is false. At least three RCT of FMT in UC are available.

R: This reviewer is correct, and we apologize for not updating our text in time. We agreed with the suggestion and performed the necessary changes, adding the following sentences with new references (last paragraph of page 13, and first paragraph of page 14, between lines 572-606):

Recently, randomized clinical trials have assessed the benefits of the use of FMT in the treatment of patients with IBD. Moayyedi et al., for example, found that patients with recently diagnosed UC could be induced to remission after treatment with FMT (2015). Data from another study showed that for patients in remission, treatment with FMT was able to maintain clinical remission in 87.1% of patients compared to 66.7% receiving placebo. These results indicate that the long-term beneficial effect of FMT in patients with UC in clinical remission could help sustain endoscopic, histological, and clinical remission (Sood et al., 2019). A recent meta-analysis showed that FMT was effective in promoting clinical remission (OR = 3.47, 95% CI = 1.93–6.25) and clinical response (OR = 2.48, 95% CI = 1.18–5.21) to patients with active UC when compared to placebo (Dang et al., 2020).

REFERENCES:

  • Moayyedi, P., Surette, M. G., Kim, P. T., Libertucci, J., Wolfe, M., Onischi, C., Armstrong, D., Marshall, J. K., Kassam, Z., Reinisch, W., & Lee, C. H. (2015). Fecal Microbiota Transplantation Induces Remission in Patients With Active Ulcerative Colitis in a Randomized Controlled Trial. Gastroenterology, 149(1), 102-109.e6. https://doi.org/10.1053/j.gastro.2015.04.001
  • Sood, A., Mahajan, R., Singh, A., Midha, V., Mehta, V., Narang, V., Singh, T., & Pannu, A. S. (2019). Role of Fecal Microbiota Transplantation forMaintenance of Remission in Patients with Ulcerative Colitis: a Pilot Study. Journal of Chron’s and Colitis. https://doi.org/10.1093/ecco-jcc/jjz060
  • Dang, X., Xu, M., Liu, D., Zhou, D., & Yang, W. (2020). Assessing the efficacy and safety of fecal microbiota transplantation and probiotic VSL#3 for active ulcerative colitis: A systematic review and meta-analysis. PLoS ONE, 15(3), 1–16. https://doi.org/10.1371/journal.pone.0228846
